# ONE PATCH TO CAPTION THEM ALL: A UNIFIED ZERO-SHOT CAPTIONING FRAMEWORK

## ABSTRACT

Zero-shot captioners are recently proposed models that utilize common-space vision-language representations to caption images without relying on paired image-text data. To caption an image, they proceed by textually decoding a text-aligned image feature, but they limit their scope to global representations and whole-image captions. We present a unified framework for zero-shot captioning that shifts from an image-centric to a patch-centric paradigm, enabling the captioning of arbitrary regions without the need of region-level supervision. Instead of relying on global image representations, we treat individual patches as atomic captioning units and aggregate them to describe arbitrary regions, from single patches to non-contiguous areas and entire images. We analyze the key ingredients that enable current latent captioners to work in our novel proposed framework. Experiments demonstrate that backbones producing meaningful, dense visual features, such as DINO, are key to achieving state-of-the-art performance in multiple region-based captioning tasks. Compared to other baselines and state-of-the-art competitors, our models achieve better performance on zero-shot dense, region-set, and a newly introduced trace captioning task, highlighting the effectiveness of patch-wise semantic representations for scalable caption generation. Code and data at: https://anonymous.4open.science/r/Patch-ioner-5E84/.

## 1 INTRODUCTION

Image captioning is one of the most representative tasks in vision-language understanding and has reached incredible accuracy thanks to the availability of pre-trained vision-language backbones and large paired image-text datasets. In its basic formulation, a captioning model takes a full image as input and autonomously decides which elements must be described and up to what degree. To enable user guidance and produce more targeted descriptions, some previous works proposed region-level captioning methods (Johnson et al., 2016; Cornia et al., 2019), which take as an additional input a spatial indication — e.g., bounding boxes — specifying which image regions have to be described and, possibly, in which order.

These region-level captioning methods require expensive manually labeled data to fully supervise the model. Indeed, each sequence or set of bounding boxes for a given image should correspond to a manually written ground-truth caption describing those objects. This fully supervised solution does not scale properly.

In this paper, we propose a perspective shift that enables us to perform region-level captioning with arbitrary spatial granularity — from a single image patch up to the entire image — in a *zero-shot* fashion, i.e., without requiring any form of image-level or region-level supervision. Specifically, instead of relying on the idea that the subject of a captioning method is the *image* — then potentially conditioned on a set of sub-regions — we instead build on two straightforward yet powerful ideas: i) the simplest element that we could caption is a *patch*, the atomic element of an image representation in modern architectures based on vision transformers (Dosovitskiy et al., 2021), and ii) we can easily aggregate multiple patch representations to produce descriptions for arbitrarily large — and also potentially not contiguous — image regions. We present a zero-shot captioning framework implementing these ideas. Our formulation offers maximum flexibility in zero-shot captioning tasks, producing models that effortlessly generate captions for various aggregations of image patches,

ranging from individual patches to larger image regions, up to providing a caption for the entire image.

Despite the powerful perspective change that defines the patch as the new captioning unit, the problem is now entangled in a simple yet critical question: *how can we craft a model able to provide patch-level captions without relying on any direct patch-level ground truth supervision?*

In the last years, large pre-trained vision-language foundation models like CLIP (Radford et al., 2021; Jia et al., 2021; Li et al., 2023a) solved many downstream tasks in zero-shot or even training-free configurations. In particular, contrastively learned vision-language representations enabled impressive results in zero-shot settings in image classification (Radford et al., 2021; Zhai et al., 2022), open-vocabulary detection (Zhou et al., 2022; Minderer et al., 2023), and segmentation (Ghiasi et al., 2022; Liang et al., 2023), or text-image retrieval (Kordopatis-Zilos et al., 2025). Image captioning, however, cannot directly employ CLIP machinery at inference time to generate text, given that CLIP is inherently a discriminative — and not a generative — approach. Only recently, image captioning models became zero-shot by decoupling image encoding — where pre-trained discriminative models like CLIP are used to create proper image and text representations — from the actual generative module. This is the case for models like Nukrai et al. (2022); Li et al. (2023b); Gu et al. (2023); Zeng et al. (2024); Fei et al. (2023); Yan et al. (2025); Zeng et al. (2025); Tewel et al. (2022); Su et al. (2022), which i) employ CLIP to leverage a shared vision-language semantic space, and ii) train a text decoder on solely text samples to recover the text back from the CLIP textual feature. This requires nothing more than a pre-trained contrastive model and a large set of sole text samples to craft a powerful captioner.

In this paper, we show that our framework can generalize this core idea and that, under this formulation, many zero-shot captioners, paired with the right components, can be easily restructured to perform zero-shot region-based captioning. Therefore, we identify and study in detail the most critical components of this novel zero-shot region-based captioning framework. Particularly, we focus our attention on the pre-trained vision-language contrastive backbone, which should be able, unlike CLIP, to create meaningful patch representations. To this aim, we largely explore DINO-based (Caron et al., 2021; Oquab et al., 2024) variants, having better localized capabilities than CLIP. We further expand the study of our framework, addressing multiple modality-gap mitigation strategies that help the text decoder to be trained only on text without having access to paired image-text features, as well as studying different patch aggregation methods.

By studying existing components and employing vision backbones able to output patch-level meaningful representations like DINO, we show that we can enable many zero-shot captioners to reach state-of-the-art or comparable results in many zero-shot captioning task variants requiring captioning sub-parts of the entire image — *dense captioning* (Johnson et al., 2016), *region-set captioning* (Cornia et al., 2019), up to the standard image captioning (Tewel et al., 2022) where the region to caption extends over the entire image. To better showcase the effectiveness of our framework in extreme patch-based captioning scenarios, we also introduce the zero-shot *trace captioning* task, which requires generating a caption for a region within an image specified by a mouse trace.

To summarize, we propose the following contributions: a) we reformulate captioning by shifting perspective from the *image-to-caption* approach to a *patch-to-caption* one, unifying local and global tasks in one framework which does not require region-level supervision, b) we repurpose existing models to work within this novel framework, by analyzing the role of the key components, with a special attention to the vision backbone, c) we show the performance of these models on four zero-shot captioning tasks that span different region granularity, from captioning few patches to the whole image, showing the effectiveness of the proposed perspective shift proposed by our framework despite its overall simplicity.

## 2 RELATED WORK

**Language-aligned Dense Image Representations** are crucial for our goal of captioning at patch level. Vision-language models (VLM) like CLIP (Radford et al., 2021) introduced a powerful approach to learning global modality representations in a shared space via contrastive learning, paving the way to solve several downstream tasks, including captioning (Mokady et al., 2021; Cornia et al., 2024). However, in zero-shot settings, CLIP-like representations are known to struggle with dense tasks

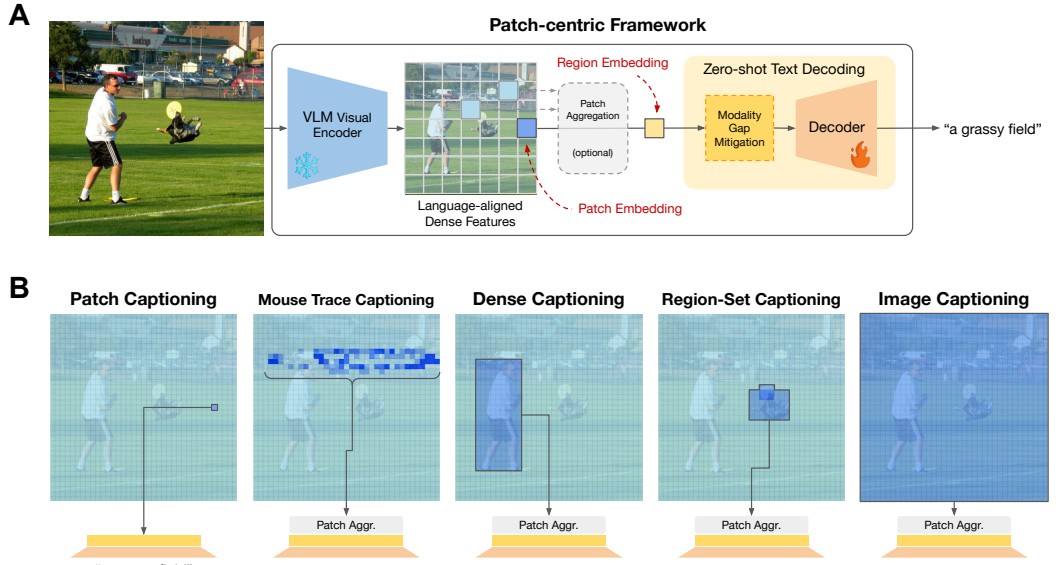

Figure 1: **Patch-centric framework for unified zero-shot captioning**. **A**. Overview of our framework. First, we extract language-aligned dense patch embeddings from the image using a VLM. Given a region, we select the underlying patches and aggregate their features to obtain a region representation. Finally, we obtain the region caption by applying a zero-shot text decoder, that is a) conditioned on the latent region representation, b) trained on text-only data, and c) equipped with a mechanism to handle the modality gap present in vision-language common spaces. This enables regional captioning without requiring region-level supervision. **B**. By aggregating patch-level features from arbitrary image regions, we can flexibly handle multiple captioning tasks across spatial granularities in a unique model.

due to misalignment between local visual patches and fine-grained semantics (Zhong et al., 2022; Ranasinghe et al., 2023; Bica et al., 2024). On the other hand, visual-only self-supervised models (SSM) like DINO (Caron et al., 2021; Oquab et al., 2024) excel in local semantic modeling but lack a bridge with language. Recent works like SILC (Naeem et al., 2024) and DINO.txt (Jose et al., 2024) attempt to get the best of both worlds by combining DINO- and CLIP-like training objectives, aiming to obtain language-aligned dense representations. INViTE (Chen et al., 2024) modifies CLIP's visual encoder by zeroing the attention weights from each patch to all the others in the last layers, leading to more semantic patch features. DenseCLIP (Rao et al., 2022) and RegionCLIP (Zhong et al., 2022) extend CLIP with additional region-level supervision. RegionCLIP leverages a region-proposal network to construct region-text pairs from image-text datasets, while DenseCLIP introduces a pixel-to-text matching loss to strengthen the alignment between local regions and textual concepts. Other methods instead exploit already existing VLMs and SSMs to get the same properties with minimal or no training: Talk2DINO (Barsellotti et al., 2024) connects language to the DINOv2 space by mapping CLIP textual representations to DINOv2 patches. ProxyCLIP (Lan et al., 2024) instead leverages DINO's attention maps to improve the local properties of the CLIP visual embeddings of patches.

**Zero-shot Image Captioning** methods rely mostly on global CLIP representations to guide text generation. *Early-guided* decoding methods take CLIP visual features as input and introduce adaptation techniques to reduce the visual-textual modality gap (Liang et al., 2022). DeCap (Li et al., 2023b) projects CLIP visual features into a more text-aligned space using a memory of texts as basis, while CapDec (Nukrai et al., 2022) and CLOSE (Gu et al., 2023) inject noise during text-only training to enable decoding also from the CLIP visual space. To improve the generation ViECap (Fei et al., 2023), MeaCap (Zeng et al., 2024), MERCap (Zeng et al., 2025) and EntroCap (Yan et al., 2025) leverage *external knowledge* to condition the decoding together with the CLIP image representation. *Late-guided* decoding methods instead use CLIP as a scoring or optimization signal rather than direct input. ZeroCap (Tewel et al., 2022) leverages CLIP gradients to steer the cached context during

text generation, while MAGIC (Su et al., 2022) optimizes token selection based on CLIP similarity scores. However, all the above approaches rely on global representations, which are not well-suited for capturing localized semantic details, making them suboptimal for patch-level or region-level captioning in zero-shot settings.

**Region-level Captioning** comprises several tasks in which models are asked to produce natural language descriptions based on sub-parts of an image. They pose additional challenges as naively captioning the cropped regions or feature maps often induces a loss of the global context of the image and, thus, misinterpretation of the region. For this reason, zero-shot solutions to this family of problems are still underexplored. For *controllable captioning* (Cornia et al., 2019) — the generation of an image caption controlled by a set or sequence of regions — and *dense captioning* (Johnson et al., 2016) — the localization and captioning of salient regions of an image — state-of-the-art solutions like CAG-Net (Yin et al., 2019), GRiT (Wu et al., 2024), ControlCap (Zhao et al., 2024), and FlexCap (Dwibedi et al., 2025) provide good performance but need supervision with ground-truth boxes. Recent works (Guo et al., 2024), (Hua et al., 2025) moved towards the direction of arbitrary regions captioning exploiting region-level supervision, and the Localized Narratives dataset Pont-Tuset et al. (2020a) — comprising images, timed captions, and timed mouse tracks — provide the ingredients for evaluating captioning also at track- or patch-level. We propose a unique framework to tackle captioning at various granularities, from image- to patch-level, in a *zero-shot setting*.

## 3 A PATCH-CENTRIC FRAMEWORK

### 3.1 MOTIVATION AND FORMULATION

**Learned region fusion requires regional data.** Traditional regional captioning (Johnson et al., 2016; Dwibedi et al., 2025; Zhao et al., 2024) follows an early injection of region specification in the model. Formally, given an image $I$ and a region $R$, the region caption $t$ is modeled as $t = \mathcal{D}(I, R)$. Such models require region-caption annotations and often use dedicated models or losses per captioning task or granularity. A step forward can be moved by introducing a formulation which disentangles image encoding and postpones region specification, defining $t = \mathcal{D}(\psi(I), R)$, where $\psi(I)$ provides a visual representation of the image independent of the region $R$, and $\mathcal{D}$ performs late region selection and text decoding. In order to avoid training the whole pipeline using region-level labels, we further decompose the decoder module $\mathcal{D}(\cdot)$ into two distinct modules: a parameter-free fixed aggregation $\text{agg}_R$ of patch-level representations, and an actual text decoder $\phi(\cdot)$ not directly conditioned on regions $R$, so that $t = \phi(\text{agg}_R(\psi(I), R))$. We will detail these two factorized components in the following paragraphs.

**Parameter-free patch aggregation.** Let $I \in \mathbb{R}^{H \times W \times 3}$ be an image split into non-overlapping patches of size $P \times P$. Each patch is encoded with a vision backbone $\psi_v$, yielding a dense grid of patch-level embeddings $V = \psi_v(I) = \{\mathbf{v}_i\} \in \mathbb{R}^{\frac{H}{P} \times \frac{W}{P} \times D}$, where $\mathbf{v}_i \in \mathbb{R}^D$. Assuming that $\psi_v(I)$ extracts this spatial grid of patch embeddings, a region definition $R$ selects a subset of patch features that we aggregate to obtain the region embedding $\mathbf{v}_R = \text{agg}_R(\psi_v(I))$. We describe this aggregation as $\mathbf{v}_S = \sum_{i \in S} w_i \mathbf{v}_i$, where $S$ is the set of indices of patches that underlie the region, and $w_i$ are aggregation weights. While several aggregation functions can be considered (e.g., uniform, gaussian, attention-based), we found that the specific choice has limited impact in regional captioning (see SM§8) and report results with mean aggregation ($w_i = 1/|S|$). Using a set operator as aggregation gives us the flexibility to aggregate arbitrary sets of patches, and thus, define regions as boxes, masks, traces, single patches, or full-image grids. Empirically, we also find that not all $\psi_v$ are suitable to extracting patch-level meaningful visual representations; transformer architectures pretrained with dense local contrastive objectives (such as DINO) are significantly more robust at the patch level, as validated in §4.2.

**Zero-shot decoding.** We train a text decoder $\phi : \mathbb{R}^D \to \mathcal{T}$ using a prefix language modeling approach, where the decoder reconstructs a caption $t \in \mathcal{T}$ from its text embedding $\psi_t(t)$. If $\psi_t(t)$ is aligned to the visual encoder $\psi_v(I)$, we could directly decode the visual embeddings using a text-only decoder trained on text-only data. In such a case, we can finally decode the region representation into a natural language caption $t = \phi(\mathbf{v}_R)$. However, the assumption that $\phi(\cdot)$ can digest features from $\psi_v(I)$ while being trained to reconstruct features from $\psi_t(t)$ is often too optimistic due to the

prominent modality gap (Liang et al., 2022). In fact, text and image representations, despite being in the same multimodal space, occupy different, separated subspaces. To obtain a decoder applicable to visual embeddings, we experimented with two mitigation strategies for this gap. The first strategy, following Li et al. (2023b), introduces a projection step at inference that maps visual features into the text subspace using a memory of text embeddings. The second, inspired by Gu et al. (2023) and Nukrai et al. (2022), trains the decoder under input perturbations, encouraging robustness so that it can directly process visual embeddings. We analyze the effects of the modality gap mitigation strategies in SM§9.

Overall, our formulation offers three main advantages: a) it is zero-shot by design, in the sense that it only requires image-level textual descriptions to be trained and does not require paired image-text samples to train the text decoder, b) any region (e.g., whole image, box, mask, free-form trace, single point) is addressed identically, supporting a modular, general-purpose regional captioning pipeline, and c) our method requires only a single forward pass of the vision backbone to extract patch features for an entire image, which can then be reused to caption multiple regions without rerunning the full pipeline for each one.

## 3.2 FROM PATCHES TO REGIONS

Building on patch-level zero-shot captioning, our framework can generate captions for arbitrary regions of an image. A region is defined as a set of patches, and its representation is obtained by averaging the embeddings of its constituent patches. This simple formulation unifies several existing region-level captioning tasks, which differ only in how the relevant patches are selected, allowing us to address them without task-specific modifications. In the following, we describe the tasks considered in our evaluation and their induced patch selections.

**Image Captioning** involves generating a single caption that describes the entire image. To achieve this, we derive a global representation $\mathbf{v}_I = \text{avg}_i(\mathbf{v}_i)$ by aggregating the feature embeddings of all patches $\{\mathbf{v}_i\}$ within the image $I$.

**Dense Captioning** requires locating salient regions in an image and generating their descriptions. Following defined evaluation protocols (Johnson et al., 2016), we focus on captioning already defined boxes, effectively removing the localization subtask, which can be tackled using additional region-proposal models. Given a bounding box $B$ and the set $S_B$ of indexes of patches that intersect with $B$, we obtain the representation of the region $\mathbf{v}_B = \text{avg}_{i \in S_B}(\mathbf{v}_i)$.

**Region-set Captioning** consists of generating a single caption for multiple regions within an image, where each region is specified by a distinct bounding box. Given an image $I$ and a set of bounding boxes $\mathfrak{B} = \{B_1, B_2, \ldots, B_K\}$, we define $S_{B_k}$ as the set of patches that intersect with the $k$-th bounding box in $\mathfrak{B}$. To represent the entire set of regions, we aggregate the feature embeddings from all selected patches across all bounding boxes, which results in a combined region-level representation

$$\mathbf{v}_{\mathfrak{B}} = \frac{1}{\sum_{B \in \mathfrak{B}} |S_B|} \sum_{B \in \mathfrak{B}} \sum_{i \in S_B} \mathbf{v}_i . \tag{1}$$

Note that if a patch appears in more than one box, it is weighted more in the average.

**Trace Captioning.** To demonstrate the flexibility of our approach, we introduce *Trace Captioning*, a novel task in which the region of interest is specified by a mouse trace $T = \{p_1, \ldots, p_L\}$ with $L$ points. Each point $p_j$ is mapped to the corresponding patch index $i_j$, yielding the sequence $S_T = [i_1, \ldots, i_L]$. The trace-level representation is then obtained by averaging the embeddings of the selected patches

$$\mathbf{v}_T = \frac{1}{L} \sum_{j=1}^{L} \mathbf{v}_{i_j} . \tag{2}$$

Unlike box-based settings, this formulation allows for free-form, user-specified regions and thus enables interactive and fine-grained localized descriptions, expanding the scope of region-level captioning.

## 4 EXPERIMENTS

In this section, we first quantitatively assess the role of the backbone in our novel framework (§4.2). Then, we measure the performance of our new formulation over state-of-the-art zero-shot captioning methods, evaluating them region-specific zero-shot tasks (§4.3). Other interesting yet less influential studies — like the role of the modality-gap mitigation strategy and the patch aggregation operator $\text{agg}_R(\cdot)$ — are available in SM §8 and §9.

To create a common ground, we first introduce the employed datasets and metrics in the following section.

### 4.1 DATASETS AND METRICS

**Trace Captioning.** We build a benchmark for Trace Captioning exploiting Localized Narratives (Pont-Tuset et al., 2020b) — a dataset in which annotators vocally described objects in images while moving the mouse pointer over the described object. The dataset provides temporal annotated voice transcriptions and mouse traces for the images of many standard captioning datasets. We took the labeled COCO (Lin et al., 2014; Chen et al., 2015) with splits defined in Karpathy & Fei-Fei (2015) and Flickr30K[1](Young et al., 2014) test subsets to build the trace captioning evaluation datasets. We split long traces and transcriptions for each image into sentences, and we discard the parts of the traces that are not temporally located between the start and the end of each sentence. We discarded noisy sentences — such as the ones describing image properties (*The image is blurred*, *the image is edited*, ...) and rewrote each sentence removing uncertainties typical of voice descriptions in a more concise and caption-like style through a few-shots prompted LLM (LLama 3 Dubey et al. (2024)). After annotation cleaning, 51 COCO images resulted without clean sentences and were discarded. Each sub-trace and relative sentence comprise an independent sample, that is, we ignore the temporal information of consecutive sentences and focus on evaluating the description of each sub-trace only. Samples and additional details are available in §10.

**Dense Captioning.** We assess the performance on dense captioning tasks following the evaluation procedure of Johnson et al. (2016), omitting the bounding box proposal and evaluating only the bounding box captioning task, using ground-truth boxes as input for the models. In addition to standard caption metrics, for this task, we also report the mAP as originally defined by Johnson et al. (2016). We use the Visual Genome (VG) v1.2 (Johnson et al., 2016; Krishna et al., 2017) and VG-COCO test splits (Li et al., 2019). The former comprises 5000 images from VG, while the latter contains 2476 images present in both VG and COCO. Both contain multiple bounding box annotations per image with descriptions.

**Region-Set Captioning** We follow the evaluation protocol of Cornia et al. (2019) that originally introduced region-set captioning. We use the Flickr30K Entities (Plummer et al., 2015) and the COCO Entities (Cornia et al., 2019) datasets. Each record comprises an image, a set of bounding boxes of variable length, and a ground-truth controlled caption. We evaluate on the test splits, comprising of images in the Karpathy & Fei-Fei (2015) test splits, that consist of 3569 and 1000 images for COCO and Flickr30k versions, respectively.

**Image Captioning** We follow the standard evaluation pipeline for zero-shot image captioning, generating captions for the 5000 images in Karpathy's COCO test split. We compare with these state-of-the-art models: DeCap (Li et al., 2023b), CLOSE (Gu et al., 2023), ZeroCap (Tewel et al., 2022), MAGIC (Su et al., 2022), ViECap (Fei et al., 2023), CapDec (Nukrai et al., 2022), EntroCap (Yan et al., 2025), MeaCap[2] (Zeng et al., 2024), and MERCap (Zeng et al., 2025). For DeCap, ViECap, and MeaCap, we also report the best results reproduced by us.

**Metrics.** All datasets used in our evaluation provide a ground-truth caption for each annotation. We therefore adopt standard captioning metrics to assess the similarity between generated and reference

---

[1]Licenses for COCO and Visual Genome are CC-BY 4.0. Flickr30k images are under the terms of Flickr's user agreement.

[2]We picked the MeaCap$_{InvLM}$, which uses a GPT-2 decoder and achieves the highest scores in the zero-shot captioning task.

| Captioning Task: (Dataset) | | Trace (COCO) | | Dense (VG v1.2) | | Region-Set (COCO Entities) | | Image (COCO) | | |
|---|---|---|---|---|---|---|---|---|---|---|
| Encoder | Backbone | C | P | C | P | C | P | C | P | CLIP-S |
| CLIP | CLIP B/16 | 10.9 | 75.0 | 10.9 | 74.2 | 41.6 | 78.8 | 42.1 | 84.0 | 66.2 |
| DenseCLIP | CLIP B/16 | 18.6 | 75.3 | 19.9 | 75.2 | 51.0 | 77.6 | 28.0 | 77.0 | 57.3 |
| INViTE | CLIP B/16 | 13.8 | 76.4 | 16.8 | 77.3 | 43.3 | 78.9 | 21.3 | 79.1 | 60.6 |
| ProxyCLIP | DINO B/8 + CLIP B/16 | 16.7 | 75.7 | 15.7 | 76.0 | 41.2 | 78.4 | 28.7 | 79.0 | 61.7 |
| ProxyCLIP | DINOv2 B/14 + CLIP B/16 | 16.5 | 75.7 | 15.5 | 76.0 | 40.6 | 78.5 | 27.4 | 78.6 | 61.0 |
| DINO.txt | DINOv2 B/14 | 23.2 | **78.8** | 23.4 | 78.5 | 91.8 | 86.3 | 67.8 | 87.2 | 70.8 |
| Talk2DINO | DINOv2 B/14 | **27.9** | 78.7 | **31.9** | **78.8** | **109.1** | **87.5** | **69.2** | **87.4** | **72.8** |

Table 1: **Vision-Language Backbones.** CIDEr (C) and RefPAC-S (P) across four captioning tasks.

captions. In particular, we focus on CIDEr (C) (Vedantam et al., 2015), which captures syntactic overlap, and RefPAC Score (P) (Sarto et al., 2023), a more recent metric that quantifies semantic similarity independently of caption phrasing. For completeness, results with other traditionally used metrics — BLEU@4 (Papineni et al., 2002), METEOR (Banerjee & Lavie, 2005), ROUGE-L (Lin, 2004), and SPICE (Anderson et al., 2016) — are reported in supplementary materials, as they follow the same trends. For the image captioning task, we additionally report CLIP-Score (Hessel et al., 2021), which measures the alignment between an image and its generated caption in the joint CLIP vision–language space. This metric is not applicable to region-set, trace, or dense captioning tasks, where captions describe local regions rather than the entire image.

## 4.2 Backbone Selection

The choice of the visual backbone is crucial for our patch-centric framework, as the quality and semantic richness of the patch features directly impact the captioning performance. We tested several state-of-the-art vision-language models pre-trained *without region-level supervision* and evaluated their effectiveness within the framework. We tested vanilla **CLIP** (Radford et al., 2021), three CLIP adaptations for dense tasks — **DenseCLIP** (Rao et al., 2022), **INViTE** (Chen et al., 2024), and **ProxyCLIP** (Lan et al., 2024) —, and two methods with visual encoders based on DINOv2 (Oquab et al., 2023) — **DINO.txt** (Jose et al., 2025) and **Talk2DINO** (Barsellotti et al., 2024).

Patches are aggregated as described in §3.2, while for the zero-shot decoder, we align with the setting of Li et al. (2023b), and use a prefix GPT-2 style decoder (SM§6 reports implementation details), with a memory-based latent projection approach as mitigation strategy for handling the modality gap. Specifically, before decoding, the region representation $\mathbf{v}$ is projected into the text embedding space as a similarity-weighted linear combination of memory elements, $\mathbf{v}_{\text{proj}} = M \alpha$ with $\alpha = \text{softmax}\left(\frac{1}{\tau} M^\top \mathbf{v}\right)$, where $M = [\mathbf{m}_1, \ldots, \mathbf{m}_N]$ stores the text embeddings $\mathbf{m}_j = \psi_t(t_j)$ and $\tau > 0$ controls the sharpness of the weighting distribution. This choice enables us to be directly comparable with other works using the same architecture in the all the following experiments, besides also performing the best among the tested zero-shot decoder methods (see SM§9). In SM§8, we also study how the choice of the memory bank $M$ affects the captioning performance, providing an upper bound to metrics.

Table 1 shows that backbone effectiveness in our framework is closely tied to capturing fine-grained local semantics. Standard CLIP performs poorly, indicating that its patch tokens lack the spatial detail needed for our tasks (Mukhoti et al., 2023; Ranasinghe et al., 2023; Bica et al., 2024). Backbones that strengthen CLIP local representations, such as INViTE and DenseCLIP, achieve stronger results, supporting our hypothesis. The best performance comes from DINOv2-based models, including DINO.txt and Talk2DINO, with the latter emerging as the most effective encoder. This underscores the importance of semantically rich patch-level features for high-quality region-level captions. For this reason, we show results using Talk2DINO as the default backbone in subsequent experiments.

## 4.3 Comparison with SOTA

Despite significant advances in zero-shot image captioning, we are unaware of any prior methods specifically tailored for zero-shot regional captioning tasks. Existing zero-shot captioners are usually evaluated only at the level of whole images, without any mechanisms to natively attend to arbitrary

regions. To rigorously quantify the benefit of our approach, in Table 2, we compare against both state-of-the-art zero-shot image captioners and adapted baselines: (i) state-of-the-art whole-image zero-shot captioners in their standard setting or applied to region crops, simulating regional captioning by isolating local content, and (ii) *region-supervised encoders*, thus outside our no-region-label setting, that leverage mask-based (AlphaCLIP, Sun et al. (2024)) or crop-based (RegionCLIP, Zhong et al. (2022)) attention coupled with the same zero-shot decoder, allowing them to attend to specific regions. This design ensures that our evaluation covers the strongest available baselines for both image-level and region-level zero-shot captioning. Note that we do not compare with large multimodal models tackling regional understanding and captioning, such as Guo et al. (2024); Hua et al. (2025), as they are outside of our assumptions on the available supervision by training on region-level data.

In addition to our strongest model (i.e., Talk2DINO with the memory-based mitigation strategy), we also report other combinations that express existing zero-shot captioning approaches (CLOSE (Gu et al., 2023), CapDec (Nukrai et al., 2022), VieCap (Fei et al., 2023), and MeaCap (Zeng et al., 2024)) but replacing the original CLIP backbone. Specifically, CLOSE and CapDec can be expressed by choosing the noise-injection mitigation strategy and using the standard decoder pipeline described in §4.2. The same applies to ViECap and MeaCap, although with the addition of extra knowledge in the text decoding step: ViECap uses external entity-aware prompts, while MeaCap leverages also structured concept retrieval from a knowledge base.

We use two datasets for each task — a COCO-derived dataset and an additional dataset such as Visual Genome (Krishna et al., 2017) or Flickr30k (Young et al., 2014). Figure 6 shows qualitative results.

**Patch-centric captioning excels in local, fine-grained tasks.** In the trace and dense captioning tasks, which emphasize local visual content, our patch-centric framework significantly outperforms all baselines across metrics. For trace captioning (Table 2, first group), our patch-based formulation outperforms whole-image captioners and crop-based adaptations. Models relying on global CLS representations fail to capture the precise objects and attributes under the trace. Even AlphaCLIP, that is a region-supervised backbone that can be naturally applied to traces, lags behind, underlying intrinsic limitations of the pretrained CLIP backbone. Dense captioning shows a similar trend (Table 2, second group), with our models outperforming baselines. For this task, we report crop-based adaptations for DeCap, ViECap, and MeaCap, as they provide a stronger baseline with respect to the same models applied to the global CLS (see Table 4 in SM). Although isolating regional content, these models discard broader contextual cues that are crucial for coherent dense descriptions.

**Patch aggregation extends seamlessy to context-aware captioning.** On region-set captioning (Table 2, third group), our models once again achieves state-of-the-art results, outperforming both zero-shot baselines and region-supervised models. Note that the region-set captioning task tends to align more closely with image-level captioning rather than strictly focusing on localized regions (see Figure 6 and Figure 7 in SM), as regions are intended to control an image-level caption[3]. Thus, global models also tend to perform well on those tasks, showing a narrower gap with respect to our new state of the art. By aggregating patch embeddings from arbitrary and possibly disjoint sets of regions, the model produces coherent and contextually rich captions that align with the collective semantics of the chosen areas. In contrast, whole-image methods, including ViECap and DeCap, cannot naturally incorporate regional cues, limiting their effectiveness in this setting. Importantly, our patch-based aggregation even surpasses AlphaCLIP, which relies on explicit mask supervision.

**Patch-centric models deliver comparable performance on whole-image captioning.** In whole-image captioning (Table 2, fourth group), our results remain competitive with the strongest zero-shot captioners but are slightly behind *dedicated* image-centric architectures such as MERCap (Zeng et al., 2025) and EntroCap (Yan et al., 2025). For this task, we report the performance of our models using an attention-based weighting, when helpful, as it usually performs marginally better than the standard average patch aggregation and thus represents the best available model for this task and reaches the smaller gap with state-of-the-art models (see SM§8). Notably, adding structured external knowledge or filtered retrieval (as in ViECap or MeaCap) on noise-based decoders improves fluency and informativeness, suggesting such modules are complementary than substitutes for strong regional semantics. However, they compare similarly to the model using the memory-based decoder.

---

[3]This is expected since the ground-truth captions in the COCO Entities dataset originate from the image-level annotations of COCO, as stated by Cornia et al. (2019).

| Model | Trace Captioning | | | | Dense Captioning | | | | | | Region-Set Captioning | | | | Image Captioning | | | | | |
|---|---|---|---|---|---|---|---|---|---|---|---|---|---|---|---|---|---|---|---|---|
| | COCO | | Flickr30k | | VG v1.2 | | | VG-COCO | | | COCO Entities | | Flickr30k Entities | | COCO | | | Flickr30k | | |
| | C | P | C | P | mAP | C | P | mAP | C | P | C | P | C | P | C | P | CLIP-S | C | P | CLIP-S |
| *Whole-image Zero-shot Captioners* | | | | | | | | | | | | | | | | | | | | |
| ZeroCap (Tewel et al., 2022) CVPR'22 | - | - | - | - | - | - | - | - | - | - | - | - | - | - | 14.6 | - | - | - | - | - |
| MAGIC (Su et al., 2022) ArXiv'22 | - | - | - | - | - | - | - | - | - | - | - | - | - | - | 49.3 | - | - | 17.5 | - | - |
| CLOSE (Gu et al., 2023) ICCV'23 | - | - | - | - | - | - | - | - | - | - | - | - | - | - | 81.2 | - | - | - | - | - |
| CapDec (Nukrai et al., 2022) EMNLP'22 | - | - | - | - | - | - | - | - | - | - | - | - | - | - | 91.8 | - | - | 35.7 | - | - |
| EntroCap (Yan et al., 2025) NEUCOM'25 | - | - | - | - | - | - | - | - | - | - | - | - | - | - | 94.3 | - | - | 41.5 | - | - |
| MERCap (Zeng et al., 2025) AAAI'25 | - | - | - | - | - | - | - | - | - | - | - | - | - | - | **96.0** | - | - | **45.6** | - | - |
| ViECap† (Fei et al., 2023) ICCV'23 | 24.3 | 74.4 | 12.0 | 68.8 | 14.90° | 26.4° | 74.3° | 15.2° | 26.6° | 74.3° | 102.7 | 85.0 | 31.8 | 74.9 | 89.7 | 88.5 | 75.6 | 29.8 | 80.6 | 70.5 |
| MeaCap† (Zeng et al., 2024) CVPR'24 | 22.5 | 74.4 | 12.6 | 69.8 | 15.01° | 28.6° | 75.1° | 16.0° | 28.9° | 75.1° | 97.9 | 85.2 | 38.6 | 76.4 | 86.0 | 88.6 | 77.8 | 40.1 | 82.8 | 73.9 |
| DeCap† (Li et al., 2023b) ICLR'23 | 20.5 | 75.3 | 11.2 | 71.0 | 17.75° | 24.6° | 77.8° | 17.8° | 24.9° | 77.7° | 95.1 | 87.4 | 39.4 | 78.8 | 87.4 | 90.6 | **79.3** | 40.0 | 84.8 | 76.3 |
| *With Region-level Supervision* | | | | | | | | | | | | | | | | | | | | |
| RegionCLIP (Zhong et al., 2022) CVPR'22 + Mem. (≃ DeCap) | - | - | - | - | 15.85 | 21.7 | 76.7 | 16.01 | 21.0 | 75.4 | - | - | - | - | 93.4 | **91.2** | 77.5 | 38.8 | 84.5 | 73.6 |
| AlphaCLIP (Sun et al., 2024) CVPR'24 + Mem. (≃ DeCap) | 21.3 | 75.4 | 11.8 | 71.0 | 14.63 | 19.1 | 73.9 | 14.82 | 19.4 | 73.8 | 95.1 | 87.4 | 39.5 | 78.8 | 89.7 | 91.1 | 78.2 | 40.9 | **85.4** | 75.0 |
| **Patch-based (Our Framework)** | | | | | | | | | | | | | | | | | | | | |
| T2D + Mem. (≃ DeCap) | 27.9 | 78.7 | 18.8 | 77.0 | **21.31** | 31.9 | **78.8** | **21.53** | **32.3** | **78.7** | 109.1 | 87.5 | **44.1** | **79.1** | 88.5◇ | 90.2◇ | 76.0◇ | 39.3◇ | 84.2◇ | 71.8◇ |
| T2D + Noise (≃ CLOSE, CapDec) | **29.3** | 78.1 | 19.3 | 75.6 | 20.26 | 26.3 | 77.0 | 20.33 | 26.4 | 76.9 | 97.5 | 85.6 | 37.1 | 76.5 | 65.5 | 86.2 | 70.9 | 27.8 | 80.8 | 67.0 |
| T2D + Noise + External knowledge (≃ ViECap) | 28.2 | 78.2 | 18.5 | 76.2 | 18.43 | 30.3 | 77.8 | 18.43 | 30.7 | 77.7 | **109.3** | 86.7 | 37.8 | 77.8 | 88.5◇ | 89.2◇ | 73.7◇ | 34.1◇ | 82.8◇ | 69.9◇ |
| T2D + Noise + Filtered knowledge (≃ MeaCap) | 27.4 | **78.8** | 20.3 | 77.3 | 18.66 | 31.9 | 78.9 | 19.43 | 32.3 | 78.7 | 104.4 | 86.9 | 42.3 | 78.6 | 83.0◇ | 89.6◇ | 74.8◇ | 39.4◇ | 84.4◇ | 71.4◇ |

†: reproduced by us. °: Model applied to image crops. ◇: attention-based weighting.

Table 2: Comparison of our patch-centric framework, using Talk2DINO (**T2D**), with state-of-the-art zero-shot captioning methods on trace, dense, region-set, and image captioning tasks. Our approach consistently outperforms whole-image and region-supervised baselines in local, fine-grained captioning tasks, while achieving competitive results on whole-image captioning. The table reports CIDEr (C), RefPAC (P), mean average precision (mAP) for dense captioning, and CLIP-Score (CLIP-S) when applicable; best and second-best results are highlighted in **bold** and underline, respectively.

| Patch | Trace | Dense | Region-Set | Image |
|---|---|---|---|---|

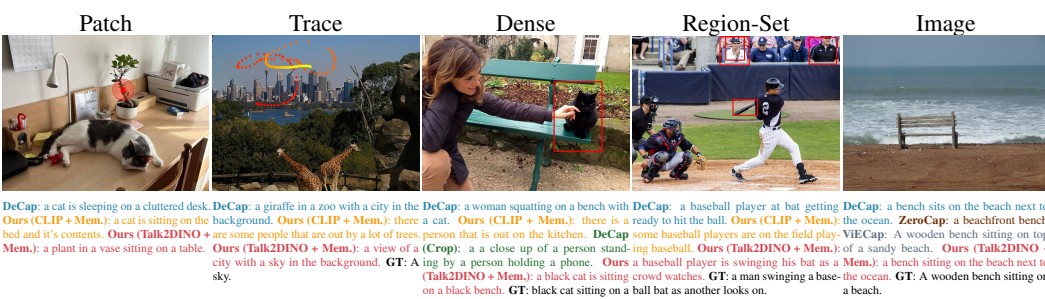

**DeCap**: a cat is sleeping on a cluttered desk. **Ours (CLIP + Mem.)**: a cat is sitting on the background. **Ours (Talk2DINO + Mem.)**: a plant in a vase sitting on a table. | **DeCap**: a giraffe in a zoo with a city in the background. **Ours (CLIP + Mem.)**: there a cat. **Ours (Talk2DINO + Mem.)**: a view of a city with a sky in the background. | **DeCap**: a woman squatting on a bench with **Ours (CLIP + Mem.)**: there is a ready to hit the ball. **(Crop)**: a a close up of a person standing. **GT**: A ing by a person holding a phone. | **DeCap**: a baseball player at bat getting **Ours (CLIP + Mem.)**: the ocean. **DeCap** some baseball players are on the field play-ing baseball. **Ours**: a baseball player is swinging his bat as a **Mem.**): a black cat is sitting crowd watches. **GT**: a man swinging a base-bat as another looks on. | **DeCap**: a bench sits on the beach next to **ZeroCap**: a beachfront bench. **ViECap**: A wooden bench sitting on top of a sandy beach. **Ours (Talk2DINO + Mem.)**: a bench sitting on the beach next to the ocean. **GT**: A wooden bench sitting on a beach.

Figure 2: **Qualitative results** from finer (left) to coarser (right) tasks. Note the discrepancy of predicted and ground-truth captions when an image-level (DeCap, DeCap (Crop)) or a CLIP-based regional (CLIP + Mem.) captioner is applied, with respect to our Talk2DINO-based model.

## 5 CONCLUSIONS

We introduced a novel zero-shot captioning framework that shifts from an image-centric to a patch-centric approach, enabling caption generation for individual patches and arbitrary aggregations without any region or even image supervision. We rely on the strong spatial awareness of DINOv2, whose local image patches have been effectively bridged with the text modality. Thanks to the disentangled training of the decoder network, this flexible and scalable method sets the new state of the art on various regional captioning tasks, including dense and region-based captioning, as well as our newly proposed trace captioning.

Despite its simplicity, results show that our patch-centric approach can effectively bridge the gap between local and global understanding in image captioning, providing a unified framework for multi-granularity captioning tasks in a zero-shot setting. Moreover, our models require a single backbone forward pass to caption multiple regions, facilitating practical viability in interactive applications.

**Limitations and Future Work.** Despite strong zero-shot performance, our model still lags behind fully supervised, task-specific approaches. The contextual scope of each patch is fixed by the backbone and is not adjusted to meet the user intents. On top of that, the modality jump introduces noise that can cause hallucinations. Future work could incorporate weak supervision, e.g., image-level captioning loss, to improve patch-level semantics in contrastively-learned representations, or refine the patch-to-text projection to further reduce the modality gap in zero-shot settings.

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

SUPPLEMENTARY MATERIAL

## 6 IMPLEMENTATION DETAILS

For the training of the textual decoder of the memory-based configuration $\phi$, we adopt a prefix GPT2-style decoder-only Transformer with 4 attention heads and 4 layers, following the architecture used by Li et al. (2023b). We train the model on captions from the COCO training set, which also serves as the memory bank $M$ for the projection mechanism, comprising approximately 500k texts. We set the hyperparameters of the projection mechanism as in DeCap ($\tau = 0.01$), and use the AdamW optimizer with a weight decay of 0.01. Training proceeds for 10 epochs with a learning rate of 10 5 and a batch size of 64. A comprehensive overview of the framework operating in the DeCap setting (with the projection mechanism as modality gap mitigation strategy) is provided in Figure 3.

For the external knowledge-based captioning models we performed a training of 15 epochs with a batch size of 80 captions on the same GPT2-style textual decoder using a learning rate of $2 \times 10^{-5}$ and a gaussian noise variance of $16 \times 10^{-3}$ to replicate the experimental settings of Fei et al. (2023) and Zeng et al. (2024).

All experiments were conducted on a single NVIDIA H100 GPU with 80GB of HBM3 memory. Training took approximately 25 minutes per epoch.

## 7 BACKBONE DETAILS

We briefly summarize here the characteristics of the vision-language models we tested in our framework.

- **CLIP** (Radford et al., 2021): A foundational model that learns a shared embedding space for images and text through contrastive learning. While being the most used model for global image-text alignment, its patch tokens are known to lack strong spatial and fine-grained semantic information. The input resolution of its training is 224 pixel.
- **DenseCLIP** (Rao et al., 2022): A fine-tuned version of CLIP that incorporates a pixel-text matching loss to enhance the model's ability to understand local regions. The official implementation input resolution is 640 pixel for the ViT-B/16 version.
- **INViTE** (Chen et al., 2024): This method modifies CLIP's vision transformer to bring patch tokens in the text space by disabling the self-attention mechanism. It employs the same visual encoder of CLIP, trained at 224 pixel input resolution.
- **ProxyCLIP** (Lan et al., 2024): A model that leverages the local understanding of a DINO backbone to improve CLIP's patch-level representations. It achieves this by replacing the attention maps in CLIP's final layer with DINO's attention maps, effectively transferring DINO's fine-grained spatial awareness to the CLIP embedding space. The DINO ViT-B/8 version was tested with images at 296 pixel resolution, while the DINOv2 ViT-B/14 at 518 pixel.
- **DINO.txt** (Jose et al., 2025): This model builds upon a frozen DINOv2 backbone, adding learnable transformer blocks on top. It is then trained with a contrastive objective against a text encoder to align both global and patch-level representations with language. The DINOv2 backbone was trained at the resolution of 518 pixel.
- **Talk2DINO** (Barsellotti et al., 2024): This model creates a bridge between the CLIP and DINOv2 embedding spaces. It trains a projection to map CLIP text embeddings into the DINOv2 patch space, using DINOv2's highly meaningful attention maps to identify and align with the most relevant patches during training. The DINOv2 backbone was trained at the resolution of 518 pixel.

## 8 ADDITIONAL RESULTS: AGGREGATION STRATEGIES, INPUT RESOLUTION, TEXT COLLECTION

We tested several patch aggregation strategies and input resolutions for our model and the other baselines.

Figure 3: **Patch-level Captioning**. Given an input image, we first extract dense patch-level representations using a vision transformer backbone. For a selected patch, we apply the projection-based mechanism introduced by Li et al. (2023b) to mitigate the modality gap and align its representation with the text embedding space. Finally, the transformed embedding is fed into a text decoder trained on a text-only corpus, generating a zero-shot caption for the patch.

**Patch Aggregation.** In cases where we are not captioning a single patch, we test different aggregation functions for merging the $\mathbf{v}_i$ in a selected set $S$ of visual patches:

a) **uniform**, the average box patch representations;

b) **gaussian**, for rectangular configurations of contiguous patches — i.e., either the full image or a bounding box; we consider a weighted average of patches representations where central patches weigh more; specifically, we assign to each patch $(a, b)$ coordinates in a uniform square grid $[-1, 1]^2$ (i.e., the top-left and bottom-right patches have $(-1, -1)$ and $(1, 1)$ coordinates, respectively), and a weight of $e^{-(a^2+b^2)}$ in the average, and

c) **attention**, a weighted average of box patches representations, with patch weights defined as the average attention map of the last layer of $\psi_v$.

**Input Resolution.** For patch-based captioning, we followed Talk2DINO Barsellotti et al. (2024) and used an input image resolution of 518x518, obtaining 37 14x14 patches per side when using the Talk2DINO backbone. The original DeCap Li et al. (2023b), ViECap Fei et al. (2023), MeaCap Zeng et al. (2024) implementations uses the CLIP B/32 backbone with 224x224 input images with 7 patches per side. We also tested with the CLIP B/16 backbone, resulting in 14 patches per side at 224x224 resolution, and with 592x592 input image size, to obtain the same number of patches as in our framework (37 per side).

We report results of these additional configurations for all baselines: DeCap, ViECap, and MeaCap. While the main paper reports only the best configuration per task and model, in this section, we report and discuss the results of all the tested configurations. We perform these tests on COCO-derived datasets and on VG v1.2 for dense captioning. We highlight the rows in the tables corresponding to the configurations reported in the main paper.

**Trace Captioning.** Table 3 reports trace captioning results. We did not apply the *gaussian* weighting scheme for this task, as the sparse discontinuous traces often do not identify a rectangular region needed to apply this scheme. We notice that a) the simple average of the trace patches provides the best performance in our framework b) as expected, using the CLIP B16 backbone, that extracts finer patches, improves over the standard CLIP B32 backbone used in the baseline methods, and c) resolutions higher than 224 only marginally improve performance for baselines in this task.

**Dense Captioning.** Table 4 reports the results of the dense captioning task. For our framework, changing the weighting strategy does not cause significant performance changes. The best baselines are the ones Region-based, which consist of applying the captioners to the CLS tokens of image crops specified by the bounding boxes (e.g., DeCap@224 Crop). The difference between the CLIP B/16 and B/32 versions is usually small or negligible.

**Region-Set Captioning.** Table 5 shows the results in the region-set captioning task on COCO Entities. The gap between zero-shot image captioners and our framework's captioners narrows in this task due to the more global nature of it, which requires the model to produce a caption for the whole image while focusing on certain regions. Also in this task, the choice of weighting strategy only marginally affects the performance of the models in our framework.

| Model | # Patches | Backbone | Input | Weighting | B | M | R | C | S | P |
|---|---|---|---|---|---|---|---|---|---|---|
| **Image-based** | | | | | | | | | | |
| DeCap@224 | 7 | CLIP B32 | CLS | - | 2.1 | 9.7 | 21.7 | 21.1 | 8.8 | 75.2 |
| DeCap@224 | 14 | CLIP B16 | CLS | - | 2.2 | 9.8 | 21.8 | 21.3 | 8.7 | 75.4 |
| DeCap@592 | 37 | CLIP B16 | CLS | - | 2.0 | 9.6 | 21.5 | 20.5 | 8.7 | 75.3 |
| ViECap@224 | 7 | CLIP B32 | CLS | - | 2.5 | 9.8 | 22.4 | 24.8 | 9.3 | 74.1 |
| ViECap@224 | 14 | CLIP B16 | CLS | - | 2.5 | 9.9 | 22.3 | 24.7 | 9.5 | 74.5 |
| ViECap@592 | 37 | CLIP B16 | CLS | - | 2.3 | 9.6 | 22.1 | 24.3 | 9.5 | 74.4 |
| MeaCap@224 | 7 | CLIP B32 | CLS | - | 2.3 | 9.4 | 21.5 | 23.1 | 8.9 | 74.2 |
| MeaCap@224 | 14 | CLIP B16 | CLS | - | 2.3 | 9.3 | 20.8 | 23.4 | 9.0 | 74.6 |
| MeaCap@592 | 37 | CLIP B16 | CLS | - | 2.2 | 9.1 | 20.3 | 22.5 | 9.0 | 74.4 |
| **Patch-based (Our Framework)** | | | | | | | | | | |
| T2D + Mem. ($\simeq$ DeCap) @518 | 37 | DINOv2 B14 | Patches | uniform | **2.5** | **10.7** | **23.2** | **27.9** | **12.6** | **78.7** |
| T2D + Mem. ($\simeq$ DeCap) @518 | 37 | DINOv2 B14 | Patches | attention | 2.4 | 10.4 | 22.7 | 27.6 | 12.0 | 78.1 |

Table 3: Trace Captioning results on COCO test set.

| Model | # Patches | Backbone | Input | Weighting | mAP | M | B | R | C | S | P |
|---|---|---|---|---|---|---|---|---|---|---|---|
| **Image-based** | | | | | | | | | | | |
| DeCap@224 | 7 | CLIP B32 | CLS | - | 0.15 | 8.40 | 0.94 | 15.61 | 19.38 | 9.38 | 73.71 |
| DeCap@224 | 14 | CLIP B16 | CLS | - | 0.14 | 8.48 | 0.95 | 15.70 | 19.11 | 9.40 | 73.94 |
| DeCap@592 | 37 | CLIP B16 | CLS | - | 0.15 | 8.37 | 0.92 | 15.67 | 18.53 | 9.26 | 73.91 |
| ViECap@224 | 7 | CLIP B32 | CLS | - | 0.13 | 8.25 | 1.02 | 16.06 | 24.18 | 9.97 | 73.03 |
| ViECap@224 | 14 | CLIP B16 | CLS | - | 0.14 | 8.30 | 1.01 | 15.86 | 23.81 | 9.91 | 73.49 |
| ViECap@592 | 37 | CLIP B16 | CLS | - | 0.14 | 8.17 | 1.00 | 15.86 | 23.26 | 9.82 | 73.35 |
| MeaCap@224 | 7 | CLIP B32 | CLS | - | 0.13 | 8.04 | 0.98 | 15.40 | 23.22 | 9.66 | 72.97 |
| MeaCap@224 | 14 | CLIP B16 | CLS | - | 0.13 | 8.01 | 1.03 | 15.15 | 23.37 | 9.49 | 73.55 |
| MeaCap@592 | 37 | CLIP B16 | CLS | - | 0.13 | 7.86 | 0.99 | 15.13 | 22.77 | 9.43 | 73.50 |
| **Region-based** | | | | | | | | | | | |
| DeCap@224 Crop | 7 | CLIP B32 | CLS | - | 0.17 | 10.03 | 1.35 | 18.20 | 23.61 | 10.90 | 77.09 |
| DeCap@224 Crop | 14 | CLIP B16 | CLS | - | 0.18 | 10.33 | 1.40 | 18.44 | 24.56 | 11.28 | 77.76 |
| DeCap@592 Crop | 37 | CLIP B16 | CLS | - | 0.14 | 8.30 | 1.05 | 16.39 | 17.20 | 7.78 | 75.47 |
| ViECap@224 Crop | 7 | CLIP B32 | CLS | - | 0.15 | 9.32 | 1.42 | 17.79 | 26.40 | 10.07 | 74.34 |
| ViECap@224 Crop | 14 | CLIP B16 | CLS | - | 0.16 | 9.59 | 1.46 | 18.03 | 27.13 | 10.43 | 75.62 |
| ViECap@592 Crop | 37 | CLIP B16 | CLS | - | 0.12 | 7.83 | 1.14 | 16.02 | 20.20 | 7.44 | 73.26 |
| MeaCap@224 Crop | 7 | CLIP B32 | CLS | - | 0.15 | 9.64 | 1.46 | 18.00 | 28.62 | 10.98 | 75.08 |
| MeaCap@224 Crop | 14 | CLIP B16 | CLS | - | 0.16 | 10.03 | 1.57 | 18.45 | 30.53 | 11.51 | 76.35 |
| MeaCap@592 Crop | 37 | CLIP B16 | CLS | - | 0.12 | 7.93 | 1.19 | 16.17 | 21.32 | 7.86 | 73.69 |
| **Patch-based (Our Framework)** | | | | | | | | | | | |
| T2D + Mem. ($\simeq$ DeCap) @518 | 37 | DINOv2 B14 | Patches | uniform | 0.21 | 10.63 | 1.36 | 18.59 | 31.94 | 15.03 | 78.82 |
| T2D + Mem. ($\simeq$ DeCap) @518 | 37 | DINOv2 B14 | Patches | gaussian | **0.22** | **10.82** | **1.43** | **18.82** | **32.80** | **15.48** | **79.14** |
| T2D + Mem. ($\simeq$ DeCap) @518 | 37 | DINOv2 B14 | Patches | attention | 0.21 | 10.31 | 1.27 | 18.17 | 30.58 | 14.72 | 78.69 |

Table 4: Dense Captioning results on VG v1.2 test set.

| Model | # Patches | Backbone | Input | Weighting | B | M | R | C | S | P |
|---|---|---|---|---|---|---|---|---|---|---|
| **Image-based** | | | | | | | | | | |
| DeCap@224 | 7 | CLIP B32 | CLS | - | 10.1 | 19.0 | 38.0 | 94.4 | 26.4 | 86.9 |
| DeCap@224 | 14 | CLIP B16 | CLS | - | 10.0 | 19.4 | 38.3 | 95.1 | 26.8 | 87.4 |
| DeCap@592 | 37 | CLIP B16 | CLS | - | 9.6 | 18.6 | 37.5 | 91.4 | 25.9 | 86.7 |
| ViECap@224 | 7 | CLIP B32 | CLS | - | 11.2 | 18.2 | 38.9 | 102.7 | 27.0 | 85.0 |
| ViECap@224 | 14 | CLIP B16 | CLS | - | 11.3 | 18.3 | 38.6 | 102.2 | 26.9 | 85.4 |
| ViECap@592 | 37 | CLIP B16 | CLS | - | 10.8 | 17.8 | 37.9 | 99.2 | 26.5 | 85.0 |
| MeaCap@224 | 7 | CLIP B32 | CLS | - | 10.4 | 17.7 | 37.0 | 97.9 | 25.9 | 85.2 |
| MeaCap@224 | 14 | CLIP B16 | CLS | - | 10.1 | 17.5 | 35.5 | 96.5 | 25.7 | 85.4 |
| MeaCap@592 | 37 | CLIP B16 | CLS | - | 9.3 | 16.9 | 34.6 | 91.1 | 25.5 | 85.0 |
| **Patch-based (Our Framework)** | | | | | | | | | | |
| T2D + Mem. ($\simeq$ DeCap) @518 | 37 | DINOv2 B14 | CLS | - | 9.1 | 16.9 | 35.0 | 89.4 | 25.4 | 85.5 |
| T2D + Mem. ($\simeq$ DeCap) @518 | 37 | DINOv2 B14 | Patches | uniform | 11.5 | 19.3 | 38.8 | 109.1 | 29.4 | 87.5 |
| T2D + Mem. ($\simeq$ DeCap) @518 | 37 | DINOv2 B14 | Patches | gaussian | **11.6** | **19.6** | **39.3** | **111.6** | **30.1** | **87.7** |
| T2D + Mem. ($\simeq$ DeCap) @518 | 37 | DINOv2 B14 | Patches | attention | 11.0 | 19.0 | 38.3 | 107.0 | 29.3 | 87.4 |

Table 5: Region-Set Captioning results for COCO Entities test set.

| Model | # Patches | Backbone | Input | Weighting | B | M | R | C | S | P |
|---|---|---|---|---|---|---|---|---|---|---|
| **Image-based** | | | | | | | | | | |
| DeCap@224 | 7 | CLIP B32 | CLS | - | 23.46 | 25.12 | 50.06 | 87.40 | 19.14 | 90.58 |
| DeCap@224 | 14 | CLIP B16 | CLS | - | **23.89** | **25.51** | **50.34** | **89.64** | **19.52** | **91.05** |
| DeCap@592 | 37 | CLIP B16 | CLS | - | 22.43 | 24.64 | 49.25 | 84.57 | 18.66 | 90.36 |
| ViECap @224 | 7 | CLIP B32 | CLS | - | 26.70 | 23.99 | 50.85 | 89.67 | 17.54 | 88.45 |
| ViECap @224 | 14 | CLIP B16 | CLS | - | 26.3 | 24.0 | 50.3 | 89.5 | 17.6 | 88.8 |
| ViECap @592 | 37 | CLIP B16 | CLS | - | 25.60 | 23.38 | 49.52 | 86.84 | 17.08 | 88.33 |
| MeaCap@224 | 7 | CLIP B32 | CLS | - | 24.57 | 23.12 | 47.68 | 86.66 | 17.27 | 88.76 |
| MeaCap@224 | 14 | CLIP B16 | CLS | - | 23.6 | 22.7 | 45.5 | 85.1 | 17.3 | 89.0 |
| MeaCap@592 | 37 | CLIP B16 | CLS | - | 22.01 | 21.87 | 44.72 | 80.84 | 16.69 | 88.41 |
| **Patch-based (Our Framework)** | | | | | | | | | | |
| T2D + Mem. ($\simeq$ DeCap) @518 | 37 | DINOv2 B14 | Patches | central patch | 15.68 | 18.46 | 40.84 | 55.53 | 12.66 | 84.26 |
| T2D + Mem. ($\simeq$ DeCap) @518 | 37 | DINOv2 B14 | Patches | uniform | 19.52 | 21.49 | 44.88 | 69.19 | 15.59 | 87.36 |
| T2D + Mem. ($\simeq$ DeCap) @518 | 37 | DINOv2 B14 | Patches | gaussian | 21.17 | 22.62 | 46.62 | 76.79 | 16.73 | 88.36 |
| T2D + Mem. ($\simeq$ DeCap) @518 | 37 | DINOv2 B14 | Patches | attention | 23.64 | 23.93 | 48.54 | 88.46 | 18.21 | 90.21 |
| T2D + Mem. ($\simeq$ DeCap) @518 GT Memory | 37 | DINOv2 B14 | CLS | - | 23.58 | 23.54 | 47.71 | 85.67 | 17.86 | 89.53 |
| T2D + Mem. ($\simeq$ DeCap) @518 GT Memory | 37 | DINOv2 B14 | Patches | attention | 25.66 | 24.77 | 49.83 | 93.87 | 19.09 | 90.70 |

Table 6: Image Captioning results on COCO test set.

**Image Captioning.** In Table 6, we report the results of standard zero-shot image captioning. In addition to the already described weighting schemes, we test one additional configuration for our framework that is *central patch*, where the decoding is applied to the central patch of the image. We can observe that the most effective strategy for the image captioning task is *attention*. This is coherent with results from Barsellotti et al. (2024), where they suggest the attention-weighted patch means to use Talk2DINO for global tasks such as image-text retrieval.

**Memory Bank.** Considering that in the memory-based model of our framework (that is similar to Li et al. (2023b)) we tackle the modality gap through a projection based on a collection of texts, we tested how much the selection of the texts in the memory bank influences performance. In Table 6, we also report the results obtained by that model when in its memory bank there are also ground-truth captions of the test set (rows marked with *GT Memory*). This provides a sort of upper bound to performance when varying the text collection used as memory. We observe that in this configuration, the performance only slightly improves (+0.5%), indicating that the model is robust to the choice of the memory bank.

## 9 MODALITY GAP: PROJECTION TO TEXTUAL SPACE VS TRAINING WITH NOISE

In this section, we quantitatively assess the performance of two state-of-the-art solutions to overcome the modality gap. In particular, we compared the configuration based on a memory bank of texts — the one introduced in §3 — with an alternative solution based on noise injection during the decoder training. Additionally, we include in our comparison a baseline with no modality gap mitigation (no mitig.), to highlight the benefits brought by each strategy.

**Training with Noise.** Various works Gu et al. (2023); Nukrai et al. (2022) proposed zero-shot image captioning solutions based on noise injection during the training of the text decoder. Through this strategy, the trained decoders are more effective in understanding semantic representations, even when those are not coming from the text modality. To implement this strategy in our framework, we trained the textual decoder on the same collection of captions as for the memory bank-based configuration. We adopted Talk2DINO Barsellotti et al. (2024) textual space for the decoder input space, which is aligned to DINOv2 Oquab et al. (2023) with registers Jose et al. (2024). Following the setting of Gu et al. (2023), we added Gaussian noise with $\sigma^2 = 0.08$ to the textual embeddings while leaving the other parameters unchanged (as defined in §6). In the next paragraphs, we report and compare the results for each task of Talk2DINO within our framework with the memory bank ($\mathcal{M}emory$) and with the training with noise ($\mathcal{N}oise$).

In Table 7, we compare the two modality gap mitigation strategies across multiple captioning tasks, and also report the performance of a baseline without any mitigation (no mitig.). The baseline consistently underperforms compared to both the $\mathcal{M}emory$ and $\mathcal{N}oise$ configurations, indicating that, like other contrastively learned image-text encoders Liang et al. (2022), Talk2DINO is also affected by the modality gap. These results highlight the importance of explicitly addressing this gap to

Table 7: **Mitigation of Modality Gap.** Comparison of Memory-based Projection (*Memory*) vs Noise-trained Decoder ($\mathcal{N}oise$) across tasks.

| Mitigation | Trace Captioning (COCO) | | | | | | Dense Captioning (VG v1.2) | | | | | | | Region-Set Captioning (COCO Entities) | | | | | | Image Captioning (COCO) | | | | | | |
|---|---|---|---|---|---|---|---|---|---|---|---|---|---|---|---|---|---|---|---|---|---|---|---|---|---|---|
| | B | M | R | C | S | P | mAP | M | B | R | C | S | P | B | M | R | C | S | P | B | M | R | C | S | P | CLIP-S |
| no mitig. | 1.2 | 9.1 | 18.3 | 14.7 | 8.5 | 75.1 | 0.18 | 9.7 | 0.7 | 15.9 | 17.8 | 10.2 | 75.2 | 5.0 | 15.0 | 29.4 | 59.4 | 21.1 | 82.2 | 9.9 | 17.7 | 36.8 | 43.7 | 12.3 | 82.2 | 69.6 |
| $\mathcal{N}oise$ | **3.0** | **11.5** | **24.7** | **29.3** | 12.3 | 78.1 | 0.20 | 10.4 | 1.2 | 17.8 | 26.3 | 12.6 | 77.0 | 10.5 | 18.4 | 37.2 | 97.5 | 26.7 | 85.6 | 19.6 | 21.5 | 45.4 | 65.5 | 15.5 | 86.2 | 70.9 |
| *Memory* | 2.5 | 10.7 | 23.2 | 27.9 | **12.6** | **78.7** | 0.21 | 10.6 | 1.4 | 18.6 | 31.9 | 15.0 | 78.8 | 11.5 | 19.3 | 38.8 | 109.1 | 29.4 | 87.5 | 19.5 | 21.5 | 44.9 | 69.2 | 15.6 | 87.4 | 72.8 |

achieve strong captioning performance. For Trace Captioning, the *Memory* method is slightly more effective in the semantic metric RefPAC-S, while the $\mathcal{N}oise$ variant achieves marginally better scores in CIDEr, ROUGE-L, METEOR, and BLEU@4, with a minimal gap between the two approaches. In Dense Captioning, the *Memory* model consistently outperforms the $\mathcal{N}oise$ model across all metrics. Similarly, for Region-Set Captioning, both methods achieve strong results, but the *Memory* method shows a clearer advantage, particularly in tasks closer to the patch level. Finally, in Image Captioning, the performance gap between the two architectures narrows, especially on the Flickr30k test split. In this scenario, the *Memory* method performs significantly better when applied to the CLS token, whereas patch aggregation produces comparable results. However, the metrics reveal conflicting trends across different datasets.

**Chosen Strategy.** Based on the observed results, we selected the projection-based approach (*Memory*) as the primary strategy for overcoming the modality gap in our framework. While the noise injection method ($\mathcal{N}oise$) yielded competitive performance across multiple tasks, the *Memory* method demonstrated superior performance in dense captioning and region-set captioning, as well as a clear advantage when applied to the CLS token in image captioning. Given these trends, and considering the stability of the projection-based approach across different evaluation settings, we adopted *Memory* as the default configuration for our framework.

## 10 TRACE CAPTIONING BENCHMARK GENERATION

We construct our Trace Captioning dataset from the Localized Narratives dataset Pont-Tuset et al. (2020a). This dataset consists of mouse traces and their corresponding speech transcriptions, where annotators describe objects in images while moving the mouse pointer over them.

The initial dataset samples include timestamped mouse traces and are composed of multiple sentences that thoroughly describe the trace, with the generated descriptions following the order of the mouse movement. However, our task does not require strict temporal coherence. Instead, we aim to generate a single, concise caption that describes the specific area covered by the localized trace, rather than a multi-sentence description.

To achieve this, we split the descriptions into individual sentences and align the traces accordingly. We then refine the traces by removing intermediate periods caused by transitions between sentences, which often occur when the annotator moves to a different region of the image. Specifically, we trim each trace by removing the first and last 15% of points, eliminating these transitional segments.

Furthermore, we refine the captions by prompting the Llama3 8B model to rephrase the sentences, removing vague or subjective phrases such as "there is," "we can see," or "on the left of the image," and replacing them with concise, objective descriptions that refer specifically to the region covered by the trace. This rephrasing is crucial to ensure that each caption adheres to the standard format of image-captioning datasets and focuses only on the precise part of the image that the trace corresponds to. The LLM also helps identify and remove irrelevant sentences (e.g., "the image is blurred," "the image is edited"), which are then discarded along with their associated traces from the final benchmark.

Figure 4 shows the full prompt used to guide the Llama model in refining and cleaning the descriptions. Figure 5 illustrates how the initial narrative samples are transformed into final trace captioning samples through the process of trace splitting and caption rephrasing.

## 11 MORE QUALITATIVE RESULTS

Additional qualitative results are shown in Figures 6 and 7. Note that the first rows of Figures 6 and 7 contain also qualitative results for single patch captioning, for which we do not have annotated data to report quantitative results.

As can be noticed in Figures 6 and 7, the Region-Set Captioning task tends to align more closely with image-level captioning rather than strictly focusing on localized regions. This is expected since the ground-truth captions in the COCO Entities dataset originate from the image-level annotations of COCO, as stated in Cornia et al. (2019).

```
1026    I have image descriptions derived from spoken narratives. These need to be
1027    ↪  rewritten as concise, stand-alone captions in the style of the image-caption
1028    ↪  datasets. Follow these rules:
1029
1030    - Remove unnecessary narrative phrases like "we can see," "there is," "in this
         ↪  image," etc.
1031    - Ensure the caption is standalone and descriptive.
1032    - Use simple, objective language that highlights key elements.
1033    - Keep it concise--just a single phrase.
1034    - Follow the classical style of caption datasets.
1035    - If the description is vague, subjective, or does not describe a concrete visual
         ↪  element (e.g., "The image is taken indoor," "This image is blurred"), return
1036    ↪  `<INVALID>`.
1037    - Wrap the output in `{}` and add nothing else.
1038
         ### **Examples:**
1039    - **Input:** "We can see a young elephant stands which is near the water in a
1040    ↪  wooded area."
1041      **Output:** {A young elephant stands near the water in a wooded area.}
1042
1043    - **Input:** "In this image I can see some young children kicking a soccer ball
         ↪  in a field."
1044      **Output:** {A group of young children kicking a soccer ball around a field.}
1045
1046    - **Input:** "In the left of the image, we see a pole that has two green street
         ↪  signs on it."
1047      **Output:** {A pole has two green street signs on it.}
1048
1049    - **Input:** "We can see two surfboards which are stuck in the sand along the
         ↪  seashore."
1050      **Output:** {Two surfboards stuck in the sand along the seashore.}
1051
1052    - **Input:** "This image consists of a man which rides a wakeboard behind a
         ↪  boat."
1053      **Output:** {A man rides a wakeboard behind a boat.}
1054
1055    - **Input:** "In the background, there are a bunch of sticky notes and a pair of
         ↪  scissors."
1056      **Output:** {A bunch of sticky notes and a pair of scissors.}
1057
1058    - **Input:** "It looks like a sepia-toned photograph of a motorcycle underneath
1059    ↪  the shadow of a
1060    tree."
         **Output:** {A sepia-toned photograph of a motorcycle underneath the shadow of
1061      ↪  a tree.}
1062
1063    - **Input:** "There is a sky"
         **Output:** {A sky.}
1064
1065    - **Input:** "She is smiling."
1066      **Output:** {A smiling girl.}
1067
1068    - **Input:** "The image is taken indoor."
         **Output:** {<INVALID>}
1069
1070    - **Input:** "This image is edited."
1071      **Output:** {<INVALID>}
1072
1073    - **Input:** "The image is blurred."
         **Output:** {<INVALID>}
1074
1075    - **Input:** "I think he is about to jump."
1076      **Output:** {<INVALID>}
1077
         Now, rewrite the following captions accordingly. Wrap each in `{}` and add
1078    ↪  nothing else:
1079    <INPUT CAPTION>
```

Figure 4: **LLM Prompt for rephrasing trace captions**.

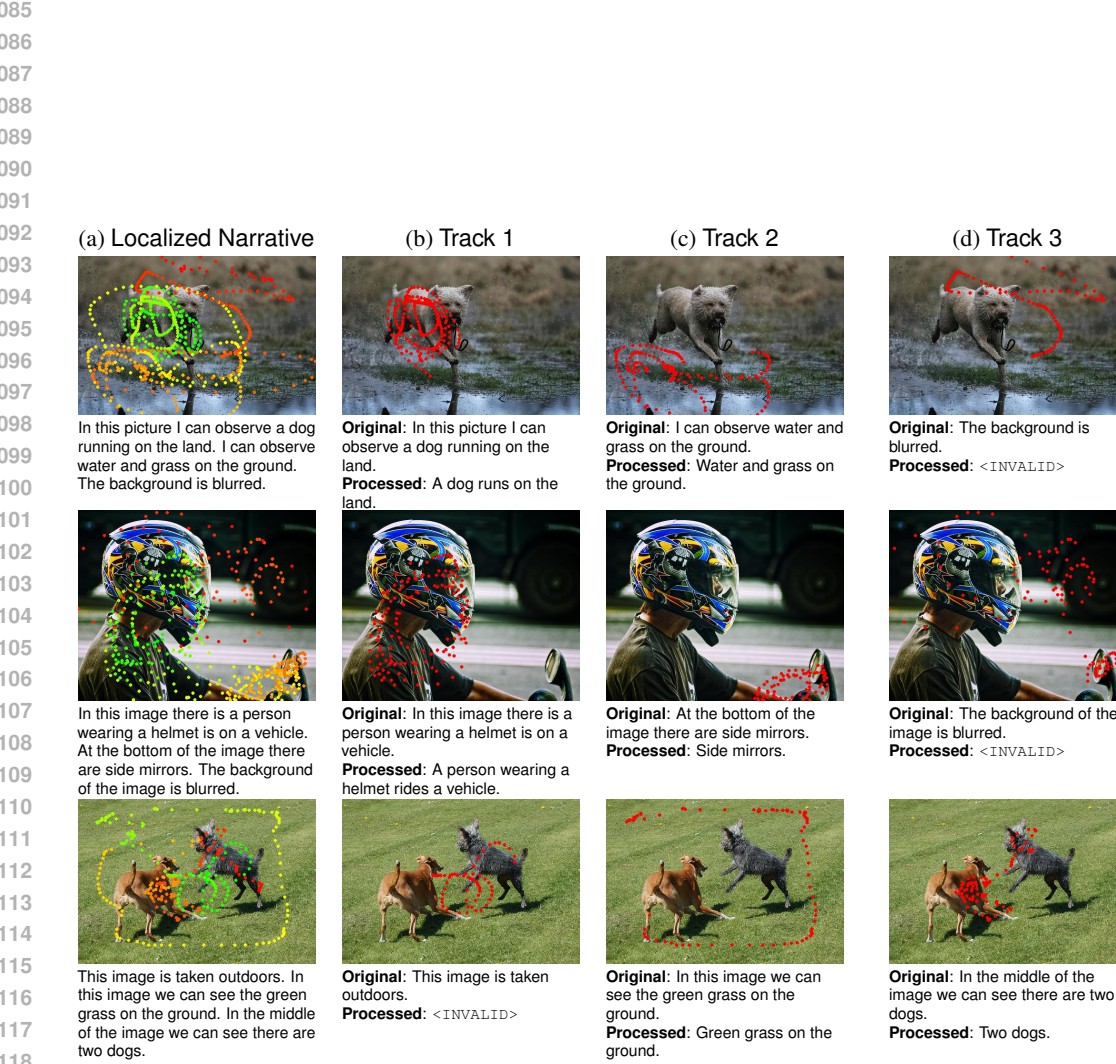

Figure 5: **Narrative vs. Trace Samples**. The first column displays sample images from the Localized Narrative dataset Pont-Tuset et al. (2020b). The remaining three columns show the corresponding mouse traces, along with the captions generated by the LLM. Captions marked with `<INVALID>` are removed from the dataset.

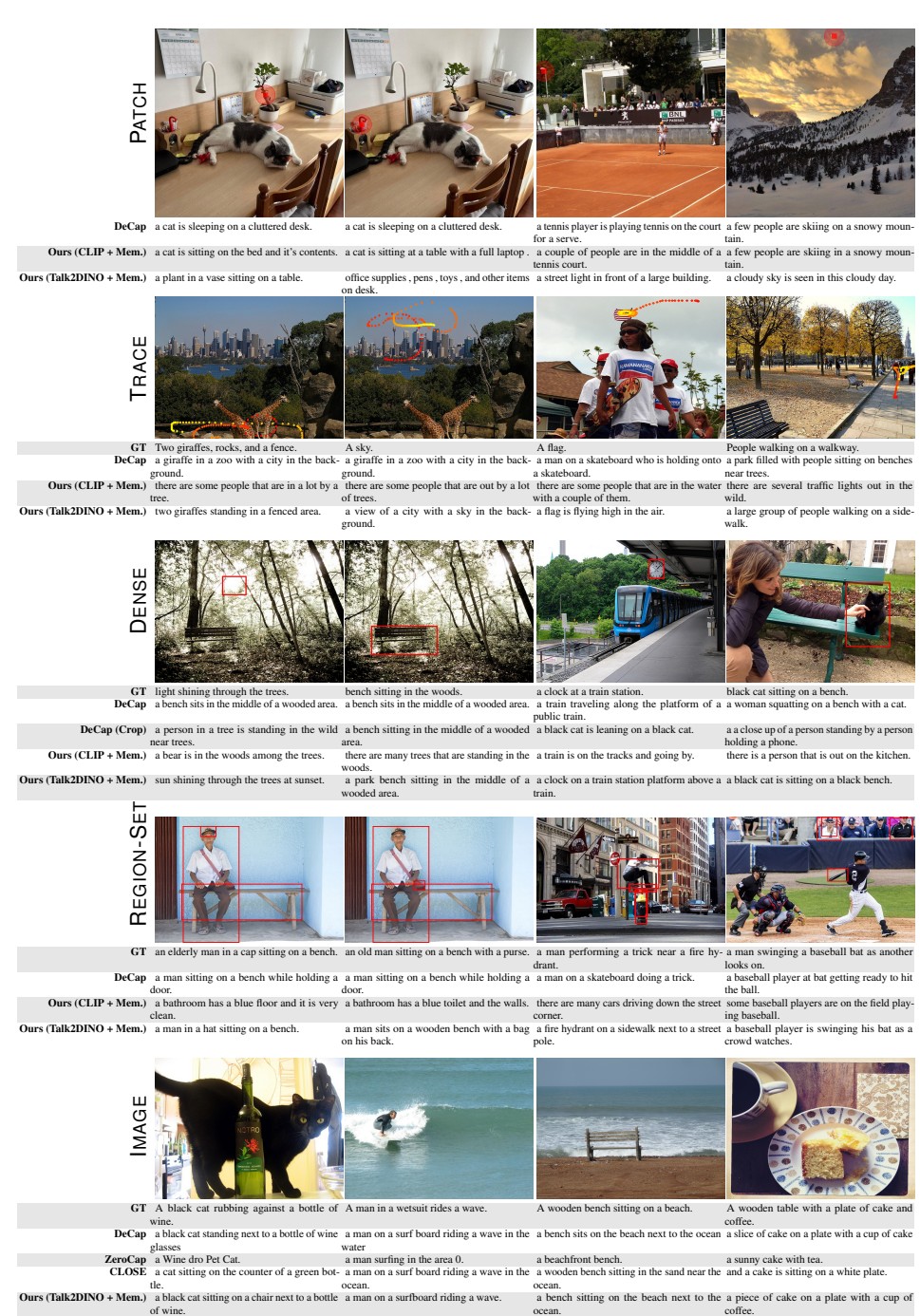

Figure 6: **Qualitative results.** We report four predictions of our model and compare baselines from the finer (top) to the coarser (bottom) task. For trace captioning examples, the trace time is color-coded from start (red) to end (yellow). **DeCap** = DeCap applied on the whole image. **DeCap (Crop)** = DeCap applied on cropped box. **ZeroCap** = ZeroCap Tewel et al. (2022) applied to the whole image. **CLOSE** = CLOSE Gu et al. (2023) applied to the whole image. **Ours (CLIP + Mem.)** = Our patch-based framework using CLIP as backbone and the projection as modality gap mitigation strategy. **Ours (Talk2DINO + Mem.)** = Our patch-based framework using Talk2DINO as backbone and the projection as modality gap mitigation strategy. **GT** = ground-truth caption.

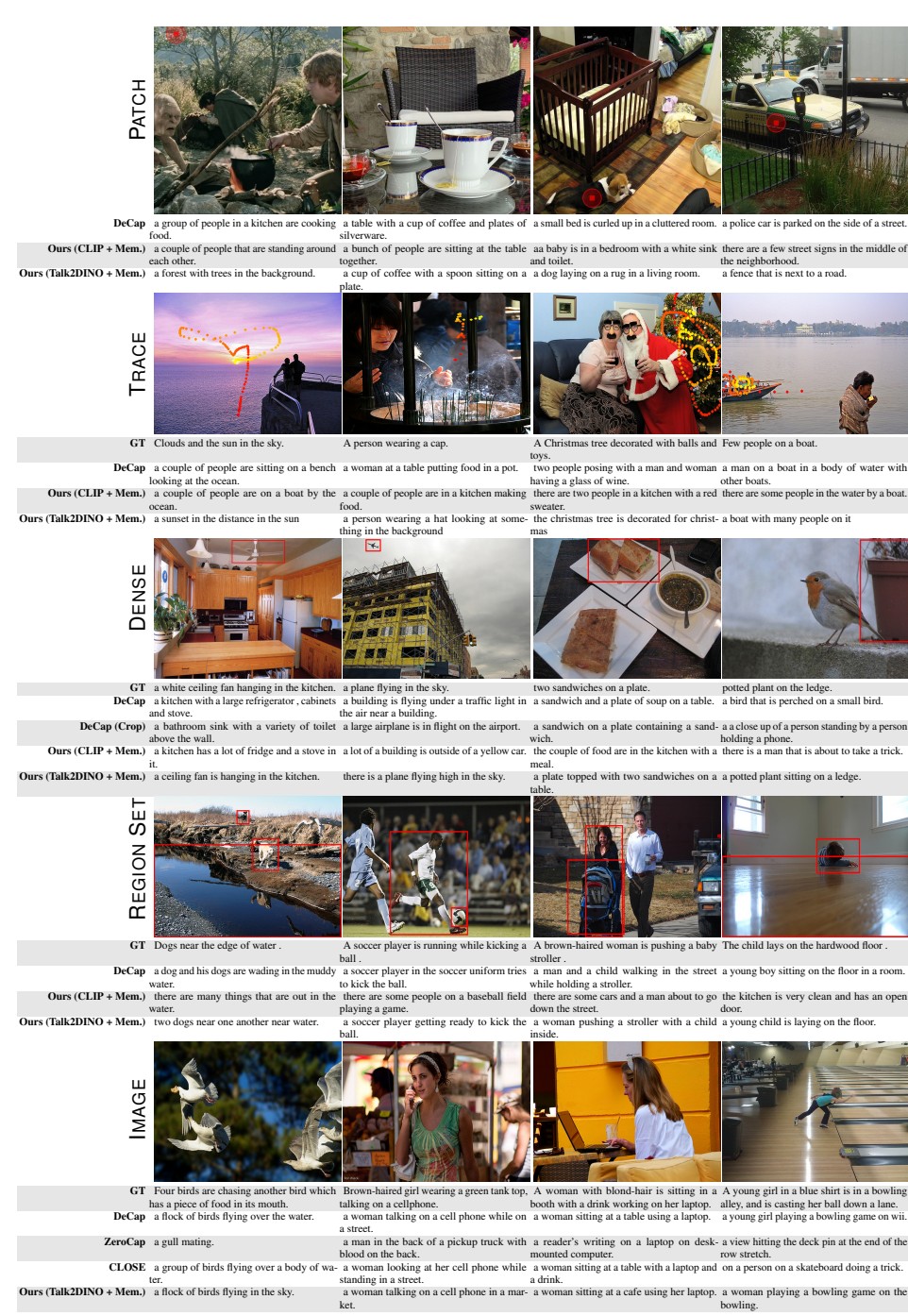

Figure 7: **Qualitative results.** We report four predictions of our model and compare baselines from the finer (top) to the coarser (bottom) task. For trace captioning examples, the trace time is color-coded from start (red) to end (yellow). **DeCap** = DeCap applied on the whole image. **DeCap (Crop)** = DeCap applied on cropped box. **ZeroCap** = ZeroCap Tewel et al. (2022) applied to the whole image. **CLOSE** = CLOSE Gu et al. (2023) applied to the whole image. **Ours (CLIP + Mem.)** = Our patch-based framework using CLIP as backbone and the projection as modality gap mitigation strategy. **Ours (Talk2DINO + Mem.)** = Our patch-based framework using Talk2DINO as backbone and the projection as modality gap mitigation strategy. **GT** = ground-truth caption.

