# OpenReview forum: "One Patch to Caption Them All: A Unified Zero-Shot Captioning Framework"
_ICLR.cc/2026/Conference — ICLR 2026 Conference Withdrawn Submission_

### Official Review · Reviewer_3nYx · 2025-10-31

**Soundness:** 3
**Presentation:** 3
**Contribution:** 3
**Rating:** 6
**Confidence:** 4

**Summary:**

This paper presents a captioning framework that adopts a patch-level approach to zero-shot image captioning, enabling universal application not only to whole-image captioning but also to various local dense captioning tasks.  The methodology leverages DINOv2-based backbones like Talk2DINO without separate supervised training. By fusing patch semantic information through simple average feature aggregation, it enables flexible information processing for arbitrary regions (entire image, specific rectangles, traces, single points, etc.). Consequently, it achieves performance improvements across diverse local-aware captioning tasks such as trace captioning, dense captioning, and region-set captioning.

**Strengths:**

- The Introduction section effectively explains the necessity of this work, proposing a training-free method that extends existing approaches limited to dense captioning and region-set captioning to enable captioning for arbitrary regions.
- To validate this method, a novel “Trace Captioning” task is proposed, and the mouse-trace and speech datasets are reprocessed to construct the required dataset for this task.
- The patch feature aggregation is achieved through a relatively simple averaging approach, making the methodology intuitive. The method also requires only a single vision backbone forward pass, resulting in low computational complexity.
- Achieves state-of-the-art performance on both trace captioning and dense captioning, which are local fine-grained tasks.

**Weaknesses:**

- Demonstrates lower performance than existing models on the whole image captioning task. This suggests that the patch aggregation method used in this study may not sufficiently capture global context.
- Had captions been generated for spatially separated regions, experiments better leveraging the characteristics of patch-level captioning could likely have been conducted. It is also regrettable that trace captioning only handles contiguous regions. While region-set captioning does include cases where separated regions are handled simultaneously, these instances are not sufficiently numerous and remain limited.

**Questions:**

- The paper does not mention the scale of the trace captioning dataset newly used in this work. How many images are in total, and what is the average caption length?
- Since the paper “Patch Matters: Training-free Fine-grained Image Caption Enhancement via Local Perception” (Peng et al.), presented at CVPR 2025, shares similar motivations and approaches to this paper, how about adding it to the citations?

---

### Official Review · Reviewer_k2Jg · 2025-10-31

**Soundness:** 2
**Presentation:** 1
**Contribution:** 2
**Rating:** 2
**Confidence:** 4

**Summary:**

This paper introduces a novel unified framework for zero-shot region-level image captioning. It treats individual image patches as atomic captioning units, enabling the generation of captions for arbitrary regions. The key to its success is the use of vision backbones like DINOv2 that provide semantically meaningful, language-aligned patch-level representations.

**Strengths:**

- Meaningful Problem: Tackling unified, zero-shot region-level captioning is a relevant and valuable research direction.
- Empirical Finding: The paper successfully identifies and demonstrates that vision-language backbones producing semantically rich, language-aligned patch-level features (like Talk2DINO) are highly effective for fine-grained captioning tasks.

**Weaknesses:**

- Clarity and Presentation:
  - The transition from the high-level motivation in the introduction to the technical details in Section 3 is somewhat abrupt. A high-level overview or a clear schematic illustrating the entire framework's workflow before diving into the mathematical formulations would significantly improve readability.
  - More critically, the paper does not explicitly state a core assumption of its framework: that the visual backbone's patch features must reside in, or be projectable into, a joint embedding space shared with the text encoder. This omission leaves the rationale for key components, like the memory-based projection, unclear and unjustified.
- Perceived Novelty:
  - The results in Table 1 strongly suggest that the performance gains are predominantly driven by the choice of a more powerful, DINOv2-based backbone (Talk2DINO). To solidly validate the novel patch-to-caption paradigm, the authors should provide a more detailed ablation or discussion separating the contribution of the backbone from the contribution of the patch-aggregation framework itself.
  - The core technical components for bridging the modality gap—memory-based projection and noise-injected decoder training—are established techniques from prior works like DeCap, CapDec. The paper would benefit from a clearer articulation of what is fundamentally novel in its approach beyond the strategic application and systematic evaluation of these techniques within a patch-based aggregation scheme.

**Questions:**

- On the Necessity of a Joint Embedding Space:
  - The requirements outlined for the visual backbone do not explicitly state that its patch features must reside in the same embedding space as these text embeddings. Could the authors clarify if they consider this joint space a strict necessity for their framework?
  - If the framework does not strictly require a joint space, the authors should provide a more detailed intuition for why the proposed operations are effective. Specifically, a clearer explanation is needed for: (1) the rationale behind performing similarity-based retrieval between features that may originate from disjoint visual and textual spaces, and (2) the underlying mechanism by which noise-injection enables a text-only trained decoder to generalize to non-textual visual inputs.
- To better contextualize the reported state-of-the-art comparisons, please include information about the visual backbones used by all baseline methods in the main comparison table (Table 2)

---

### Official Review · Reviewer_JXfF · 2025-11-01

**Soundness:** 2
**Presentation:** 3
**Contribution:** 2
**Rating:** 2
**Confidence:** 5

**Summary:**

This paper proposes a patch-to-caption framework for achieving regional image captioning in a zero-shot setting. The core contribution lies in shifting captioning from the image level to the patch level and treating it as a text-only decoding task, thereby eliminating the need for region-level annotations. The authors demonstrate that the proposed method is effective in specific comparisons (i.e., against certain zero-shot captioning methods).

**Strengths:**

1. The proposed method is easy to implement and follow.
2. No region-level annotation is required.
3. Performance is good under specific settings.

**Weaknesses:**

1. My main concern is the experimental setting. In #385 – 387, “we do not compare with large multimodal models tackling regional understanding and captioning… by training on region-level data.” I understand that the authors did not compare with region-level MLLMs, but I question why they did not compare with general MLLMs. I believe that there are many recent MLLMs trained only on whole image–text pairs. The authors should try to directly crop the corresponding region from the whole image and feed the cropped image to the MLLM for comparison, as these models would also be zero-shot since they were not trained on region-level image–text pairs.
2. The technical contribution is minimal. Text-only decoding has been proposed in previous methods such as DeCap and ViECap. Patching and aggregating region features is essentially a variant of cropping an image region and encoding it (as a single large patch, similar to the proposed patching and aggregating region features).
3. “We are unaware of any prior methods specifically tailored for zero-shot regional captioning tasks.” I think there is little difference between this paper and works like Caption Anything. These works first crop a specific region based on a bbox, point, or mouse trace and input it into a VLM. If this VLM has not been trained on region-level image–text pairs, it can also be considered a zero-shot regional captioning approach, right?
4. The backbone is outdated. There are much more powerful backbones available now, such as SigLIP and SigLIP2. For open-source language models, examples include Qwen-2 and Qwen-3. The authors should demonstrate the effectiveness of their method across diverse backbones.

**Questions:**

1. #198 - 199: Each path will be encoded into a single feature, i.e., 1xD?
2. #201 - 202: Vr=Vs? The authors should clarify this further.

---

### Official Review · Reviewer_MswA · 2025-11-01

**Soundness:** 4
**Presentation:** 3
**Contribution:** 2
**Rating:** 6
**Confidence:** 4

**Summary:**

This paper introduces a new image captioning framework working on a patch-based inputs instead of a whole image. It treats a single patch to a basic unit for image captioning, and can take arbitrary region from an image by aggregating the patches related to that region. This framework consists of image encoder, language decoder, and a patch aggregation mechanism in addition to a modality gap mitigation strategy. Specifically, given a region, the underlying patches are aggregated to obtain a region embedding, and this is passed to the language decoder. By its zero-shot captioning design, it only requires textual descriptions during training and even o requires region-level supervision. During inference, the patches of interest is aggregated and projected to the language space and then be fed into the language decoder. Experiments show that the proposed patch-centric captioning framework supports various image captioning tasks in literature, including region-level and global-level captioning, while achieving superior performances across these tasks. In addition, a new trace-level captioning task is proposed, where the captioner generates descriptions based on the regions traced by a mouse pointer.

**Strengths:**

A single image captioning framework that can be seamlessly compatible with different types of image captioning tasks, such as region-level and global-level image captioning and dense captioning is acknowledgeable. Shifting the perspective from image-centric to patch-centric image inputs could lead to this flexibility, which can be considered one of the main contributions. In addition, with the aid of strong image encoder and language decoder, the training process has several advantages. Under the zero-shot captioning framework, its text-only training nature removes the need for input images and even region-level supervision, reducing its training complexity.

**Weaknesses:**

- The main limitation lies in the narrow scope of the training data, as both training and evaluation are based solely on COCO and COCO-related data, where overfitting to the specific image domain and caption style could benefit to the performance (e.g., the main evaluation metrics are reference-based methods, by comparing the predicted and ground-truth captions). Verifying whether the proposed framework can generalize well to datasets with different captioning styles is expected. Moreover, since the model is trained only on COCO, it mainly covers common image domains. Therefore, it is expected to validate the framework on additional training datasets, such as more challenging or specialized domains.

- From the experiments, it can be observed that the model's performance largely depends on the specific choice of image encoder, Talk2DINO. Despite sharing the same backbone as DINO, it is necessary to present any reasoning or analysis for such a gap.

**Questions:**

Overall, the proposed patch-centric perspective is novel and offers several practical advantages, including unified support for different types of captioning tasks and relatively low training complexity. However, the framework appears to be dependent on the specific training data and choice of image encoder for its superior performance, which requires further justification.

---

### Author Response · Authors · 2025-11-14

We sincerely thank the reviewers for their time and constructive feedback. We are encouraged that the reviewers recognize the novelty of our patch-centric perspective, the flexibility of our unified framework across various captioning tasks, and the practical advantage of enabling zero-shot region-level captioning without region-level supervision or multiple backbone calls.

We acknowledge the critical feedback. Given the current evaluation, we have decided to withdraw the submission from ICLR 2026 to perform revisions and expand our empirical validation. We feel some core aspects of our methodological novelty and the zero-shot setting were misunderstood, and we use this response to clarify them.

## 1. On Technical Contribution and Novelty
The novelty of our work is not solely in inventing new components (like text-only decoding or memory projection, which we explicitly cite as existing techniques ) but in the patch-centric formulation itself, which enables a new, unified paradigm for zero-shot region-level captioning:

- Shifting the Paradigm: We shift the atomic unit from the whole image to the individual patch.
- Arbitrary Region Support: A region is defined as an arbitrary aggregation of these existing patch features (e.g., mean). This is fast, highly flexible (natively supporting non-contiguous regions), and is the key factor enabling superior performance on fine-grained tasks like Trace and Dense Captioning.
- Efficiency vs. Cropping: Our method is fundamentally different from repeatedly cropping the image and encoding the result. Our method requires only one vision backbone forward pass per image, followed by fast feature aggregation. In contrast, cropping necessitates a full, slow forward pass for every region. For instance, dense captioning datasets like Visual Genome can feature up to 100 regions per image, which would make the image encoding phase approximately 100 times slower than with our approach. Furthermore, the cropping method inherently discards the broader image context, which can lead to suboptimal results.

## 2. On the definition of Zero-shot
We acknowledge that the term "zero-shot" can cause confusion in understanding assumptions of our work due to its polysemy in the literature. We will clarify that our framework is strictly governed by two core constraints that enable unified and flexible regional captioning:
- Decoder Zero-Shot (No Paired Image-Text Training): The text decoder is trained exclusively on text samples and is never exposed to images or paired image-text data during its training (aligning with models like DeCap, ZeroCap, and ViECap). This excludes most large end-to-end VLMs.
- No Region-Level Supervision: The entire pipeline is constructed and trained without requiring any form of explicit region-level annotations (e.g., bounding boxes or masks).

By adhering to both constraints, our method provides a uniquely strict definition of zero-shot regional captioning that maximizes flexibility and minimizes data requirements. As the reviewers correctly noted, models exist (such as some VLLMs) that break Constraint 2 due to their region-level supervision, making direct comparison incorrect. However, we note that almost every other VLLM we are aware of breaks Constraint 1 by being trained end-to-end on massive paired image-text datasets, placing them outside of our setting as well. For better context and comparison against the state-of-the-art in supervised scenarios, we will consider adding VLLM comparison points as supervised baseline.

## 3.  Dependency on the visual backbone
Our paper's finding is that a semantically rich patch-based backbone is essential (Table 1). This is not a weakness but a key empirical result that validates our patch-centric premise. We perform detailed experiments (Table 1) to compare multiple backbones (CLIP, DenseCLIP, INVITE, ProxyCLIP, DINO.txt, Talk2DINO) and demonstrate why DINOv2-based models, with their superior localized features, are required for high-quality regional tasks.

---

### Note · Authors · 2025-11-14

I have read and agree with the venue's withdrawal policy on behalf of myself and my co-authors.